# Phase separation driven by interchangeable properties in the intrinsically disordered regions of protein paralogs

Shih-Hui Chiu[1], Wen-Lin Ho[1], Yung-Chen Sun[1], Jean-Cheng Kuo[1] & Jie-rong Huang [1,2,3✉]

Paralogs, arising from gene duplications, increase the functional diversity of proteins. Protein functions in paralog families have been extensively studied, but little is known about the roles that intrinsically disordered regions (IDRs) play in their paralogs. Without a folded structure to restrain them, IDRs mutate more diversely along with evolution. However, how the diversity of IDRs in a paralog family affects their functions is unexplored. Using the RNA-binding protein Musashi family as an example, we applied multiple structural techniques and phylogenetic analysis to show how members in a paralog family have evolved their IDRs to different physicochemical properties but converge to the same function. In this example, the lower prion-like tendency of Musashi-1's IDRs, rather than Musashi-2's, is compensated by its higher α-helical propensity to assist their assembly. Our work suggests that, no matter how diverse they become, IDRs could evolve different traits to a converged function, such as liquid-liquid phase separation.

[1] Institute of Biochemistry and Molecular Biology, National Yang Ming Chiao Tung University, No. 155 Section 2, Li-nong Street, Taipei, Taiwan. [2] Institute of Biomedical Informatics, National Yang Ming Chiao Tung University, No. 155 Section 2, Li-nong Street, Taipei, Taiwan. [3] Department of Life Sciences and Institute of Genome Sciences, National Yang Ming Chiao Tung University, No. 155 Section 2, Li-nong Street, Taipei, Taiwan. ✉email: jierongh@nycu.edu.tw

Recurrent gene duplications over many generations create paralogs in a genome, giving rise to a myriad of new protein functions[1]. The redundant copy, if not silenced as a pseudogene (the non-functional copy from its ancestor), enhances survival, and many duplicated genes mutate toward new functions[2]. The hemoglobin, composing paralog α- and β-globin chains, is a textbook example[3]: The existence of two copies of the α-globin gene (HBA1 and HBA2) reduces the effect of α-thalassemia, but not the β-thalassemia, which results from the loss of the only one β-globin encoding gene (HBB); These paralogs, which have evolved from an ancestral oxygen-binding globin, collaborate in their new function of oxygen transport[3]. Although protein paralogs have been studied for decades, the paralogs of intrinsically disordered proteins (IDPs) or proteins with intrinsically disordered regions (IDRs), account for more than half of the eukaryotic proteome[4], remain largely ignored.

Our particular interest is in RNA-binding proteins (RBPs), which are at the core of gene regulation and RNA metabolism and whose dysfunction is implicated in many diseases[5]. In addition to the common feature of having RNA interacting motifs[6], RBPs often contain IDRs[7,8]. The role of these IDRs had been unclear until it was recently shown that they can organize cellular structures without a lipid membrane[9]. These membraneless organelles, which assemble via liquid-liquid phase separation (LLPS), have since been extensively studied and have become a working model of spatiotemporal control for many cellular functions[10]. Although some RBPs' IDRs have been reported involved in forming membraneless organelles (such as RNA or stress granules), a large proportion of them are still of unknown function. The RBPs themselves are often paralogs, many of which share highly conserved RNA-binding domains but with different IDRs. In the manually curated list of RBPs (1542 RBPs in total)[11], more than 22% (341/1542) are annotated in one of the paralog families in the OrthoMCL database[12], and more than 48% are paralogs (749/1542) as defined in the original census study based on sequence similarity[11] (see Supplementary Data 1 and Supplementary Data 2 for the lists). These paralogs have similar folded domains, hinting at similar RNA recognition mechanisms, whereas many disordered regions have greater sequence diversity.

To understand the different properties of IDRs in paralogs, we focus on those families of proteins that have two members and found that the Musashi protein family is a suitable example for this purpose. The Musashi gene was originally identified in the fruit fly (Drosophila melanogaster), responsible for sensory organ development[13] and is highly conserved in animals[14–18]. In mammals, the two Musashi paralogs are translational regulators of cell fate and are involved in maintaining the stem-cell state, differentiation, and tumorigenesis[19]. In addition to their role in neural stem-cell development[17,18,20], they also regulate several types of cancer[21,22]. The C-terminal domain (an IDR) is critical for forming chemoresistant stress granules in glioblastoma[23] and colorectal cancer cells[24]. Musashi proteins cannot join stress granules without it[23,24]. Furthermore, the toxic oligomers formed by Musashi proteins have been implicated in Alzheimer's disease[25,26]. We noticed through sequence analysis that the IDRs of Musashi-1 and −2 have different physical properties, which are conserved among vertebrate orthologs. Here, our data suggest that the decreased prion-likeness (the level of amino acid composition resembling that of prion proteins) of one paralog is compensated in assembly formation by an increase in α-helical propensity. We also compare the IDRs' properties of other well-studied RBPs related to Musashi proteins, including TDP-43 and hnRNP A1. These results show how different properties may have evolved in IDR paralogs for the same biophysical mechanism.

## Results

### The IDRs of the Musashi family have different prion propensities.

We used the primary sequence of the folded domain of fruit fly Musashi (residues 29–195, a predicted RNA recognition motif (RRM)) to identify orthologs in the UniProt database (i.e. ignoring its IDR). We selected the model organisms[27] (listed in Supplementary Table 1) with similar RRM sequences and constructed a phylogenetic tree using their full-length sequences (Fig. 1a). The nematode and fruit fly have only one Musashi gene, but vertebrates have two paralogs (Fig. 1a). The C-terminal half of all these proteins are intrinsically disordered (purple bars in Fig. 1a and Supplementary Fig. 1). Note that the IDR was not used to identify orthologs but is a common feature of all Musashi proteins. Results suggest the IDR is involved in Musashi-1 joining stress granules[23,24]. The stress-granule-related proteins often possess a prion-like domain[28,29], which is similar to prion behavior to assist assembly but does not necessarily aggregate. We, therefore, used the PLAAC algorithm[30] to predict prion-likeness. Although all the IDRs of Musashi-2 in the vertebrates and lower animals are prion-like, the IDRs of Musashi-1 orthologs are less so (Fig. 1a). Figure 1b compares the human Musashi paralogs, where the IDRs cover residues 237–362 (Msi-1C) and 235–328 (Msi-2C) (red box). We separated all Musashi protein sequences into IDRs and RRMs based on their alignment to the corresponding human ortholog (Fig. 1b and Supplementary Fig. 2). The RRMs in Musashi-1 and Musashi-2 have very similar amino acid compositions (Fig. 1c, d and Supplementary Table 2). On the contrary, although the IDRs of the two paralogs have many glycines, prolines, and alanines (Fig. 1c and Supplementary Table 2), their amino acid sequences differ substantially (Fig. 1d, lower panel). Sorting the amino acids in terms of prion-forming propensity[30] reveals that the Musashi-1 orthologs have fewer prion-promoting amino acids, such as glutamine and asparagine than Musashi-2 orthologs do. This amino acid composition analysis reinforces the suggestion that Musashi-2 is more prion-like (Fig. 1d).

### The disordered regions of Musashi proteins have a polyalanine region that forms an α-helix.

We purified the human Musashi proteins' IDRs to investigate their differences experimentally (Supplementary Fig. 3). Their circular dichroism (CD) spectra show that they are mostly unstructured (Fig. 2a). The CD patterns of Msi-1C and Msi-2C at pH 5.5, 283 K (the conditions at which the proteins were most stable and soluble, see below), are similar overall but differ slightly around 220 nm, hinting at a potential difference in α-helicity[31,32]. We assigned the NMR chemical shifts of the two IDRs to obtain residue-specific structural propensities (BMRB accesses: 51207 and 51208). In the $^{15}$N-edited heteronuclear single-quantum coherence (HSQC) spectra (Fig. 2b), most of the amide proton signals are within 1 ppm, confirming the disordered nature of these domains[33]. Nevertheless, both regions contain a stretch of residues in which the secondary chemical shifts (the differences between the measured chemical shifts and random-coil values) are positive for Cα and C' atoms and negative for Cβ atoms, indicating a propensity of an α-helix (Fig. 2c). The deviations from random-coil values are smaller for Msi-2C than for Msi-1C (Fig. 2c). δ2D predictions[34] based only on chemical shifts (with no missing assignments in the α-helical region) indicate that the α-helical propensity of Msi-1C is higher than Msi-2C's (residue-specific values up to ~50% vs up to ~20%; red bars in Fig. 2d). These α-helical forming regions correspond to the polyalanine stretches of the two proteins (Fig. 2d, e). Msi-1C contains an eight-alanine-repeat whereas Msi-2C's polyalanine stretch is interrupted (and its α-helical propensity reduced) by a valine[35,36]. Importantly, the

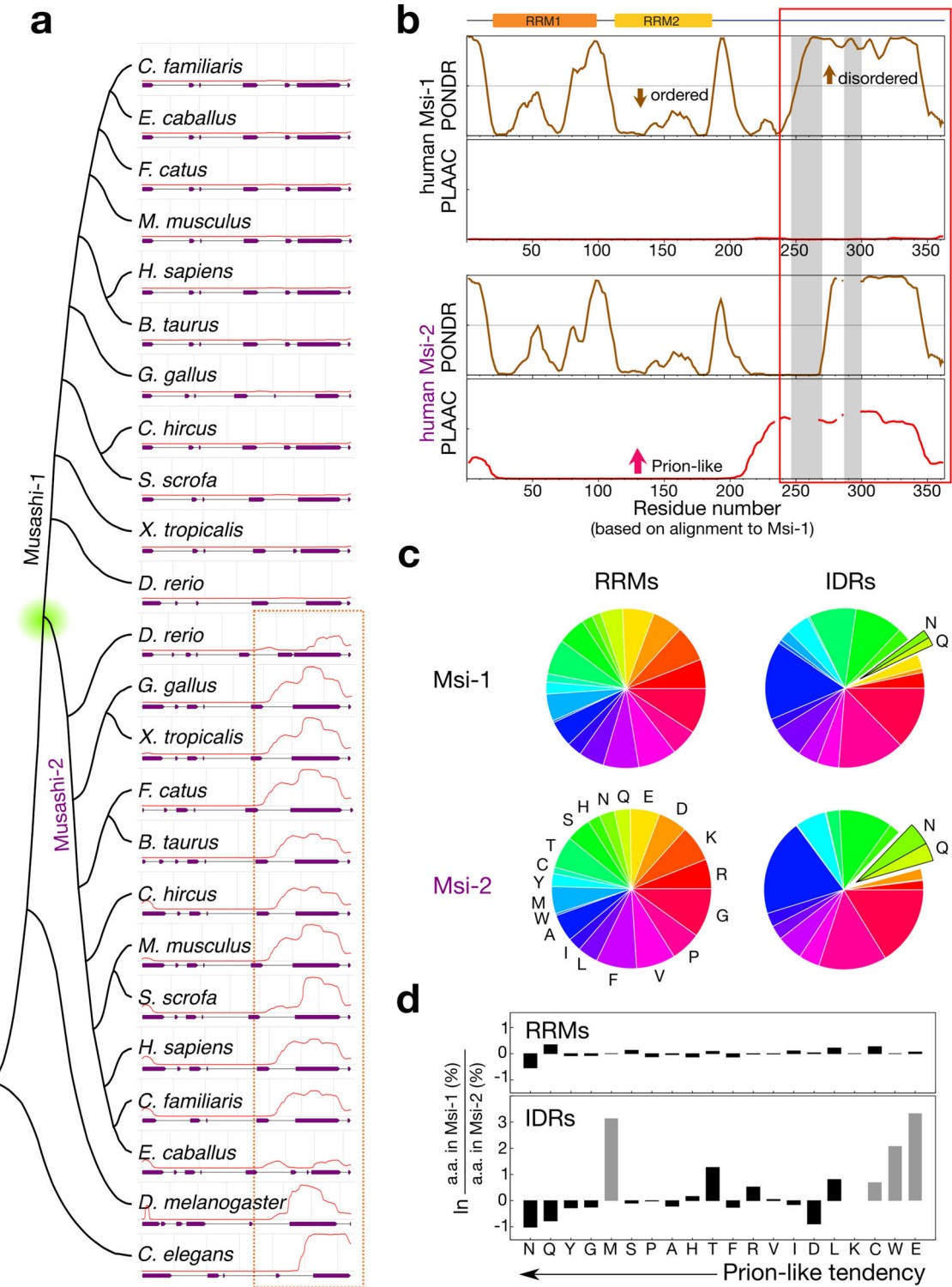

**Fig. 1 Prion-like nature of Musashi protein homologs. a** Phylogenetic tree (constructed using MEGA X[62]) of Musashi proteins in model organisms. The red lines indicate the PLAAC prion propensity scores[30]. The purple bars indicate regions predicted to be disordered by the VSL2 algorithm in PONDR[61]. The dashed box highlights the most prion-like regions. **b** Disorder and prion-likeness analysis of human Musashi proteins. Residues are numbered according to the alignment with Musashi-1 and the gray shading indicates the regions that are missing in Musashi-2. The red box contains the consensus intrinsically disordered regions (residues 237–362 of Musashi-1 and 235–328 of Musashi-2) considered in this study. **c** Amino acid pie charts of Musashi proteins in the model organisms listed in panel (**a**). The RRMs and IDRs were defined as shown in panel (**b**). **d** Single amino acid population differences in Musashi-1 and -2 with the amino acids sorted by prion-likeness[30]. Gray bars indicate amino acids that appear fewer than ten times in the sequence.

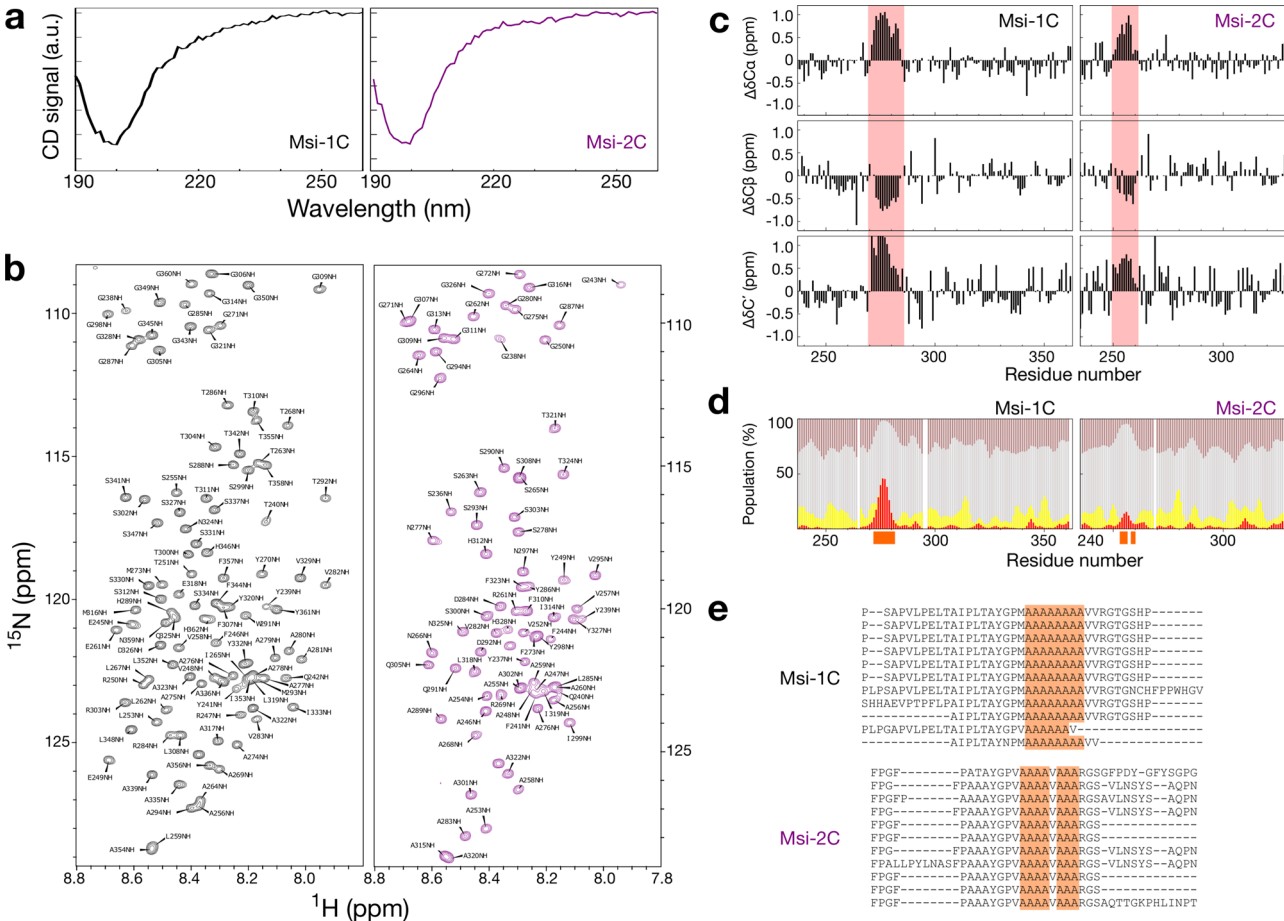

**Fig. 2 The C-terminal domains of Musashi proteins are intrinsically disordered and have helix-forming polyalanine tracts. a** Circular dichroism (CD) spectra of the C-terminal domains of Musashi-1 (black; Msi-1C) and Musashi-2 (purple; Msi-2C). **b** $^{15}$N-edited NMR HSQC spectra of Msi-1C and Msi-2C with resonance assignments. **c** Secondary chemical shifts of Cα, Cβ, and carbonyl-carbon (C′). The most pronounced deviations are highlighted in red. **d** Stacked plots of the secondary structural populations derived from the chemical shifts using the δ2D algorithm[34]. Red: α-helix; yellow: β-sheet; gray: random coil; brown: polyproline II helix. The alanines in the polyalanine tracts are indicated by orange blocks on the x-axis. **e** Multiple sequence alignment of Musashi homologs. The sequences are in the same order as shown in Fig. 1a. The polyalanine stretch is highlighted in orange. The CD and HSQC spectra were repeated at least three times for both protein samples.

respectively continuous and disrupted polyalanine tracts are conserved in Musashi-1 and Musashi-2 orthologs (Fig. 2e and Supplementary Fig. 2).

**The non-conserved regions flanking the polyalanine tracts tune their α-helical propensity.** Musashi-1 has two additional regions not found in Musashi-2 around the polyalanine tract (denoted Seq1 and Seq2; Fig. 3a). We created three constructs without Seq1, Seq2, or both, to investigate their effect (Fig. 3b; ΔSeq1, ΔSeq2, and ΔSeq1ΔSeq2). The CD patterns of these variants are similar to those of Msi-1C (Fig. 3c) and in the $^{15}$N-$^1$H HSQC spectra, most peaks overlap with those of Msi-1C. However, there are pronounced changes between Seq1 and Seq2 (Fig. 3d). The adjacency of these regions to the mutation sites is not the only cause because residues further away from the polyalanine region show smaller chemical shift perturbations than those within the polyalanine region (as highlighted in Fig. 3d as an example and quantified in Fig. 3e). These changes show systematic patterns with downfield chemical shifts in the proton and nitrogen dimensions for ΔSeq1 and the opposite trend for ΔSeq2 (Fig. 3d, e), indicating a shifting equilibrium between different conformations in the fast exchange regime. To confirm this, we assigned the $^{13}$C chemical shifts of the three variants (BMRB accesses: 51204, 51205, and 51206; Supplementary Fig. 4a), which

are mostly similar, except those close to the polyalanine region (Supplementary Fig. 4b). We calculated the secondary chemical shift difference of Cα and Cβ atoms (ΔδCα-ΔδCβ; which minimizes the error from chemical shift referencing) and compared these values in the variants to those in the wildtype (Δ(ΔδCα-ΔδCβ); Fig. 3f). Since larger (ΔδCα-ΔδCβ) values indicate a stronger α-helical tendency, these results indicate that the wildtype's α-helical propensity is lower than ΔSeq1's but higher than ΔSeq2's. The estimated α-helical propensities (using the δ2D program) are up to 6% higher than in the wildtype for ΔSeq1 but up to 5% lower than in the wildtype for ΔSeq2 (Fig. 3g). These results are in keeping with the indistinguishable CD data because, relative to the entire length of the sequences, the difference in α-helical propensity is negligible (<10% difference over just 10% of the sequence). The ΔSeq1ΔSeq2 variant has a similar predicted secondary structural population as the wildtype (Fig. 3g). These results indicate that α-helical propensity around the polyalanine stretch is altered by these non-conserved variations in primary sequence with, in order of α-helical propensity, ΔSeq1 > wildtype ≈ ΔSeq1ΔSeq2 > ΔSeq2. Both these regions modify the α-helical propensity probably by altering the equilibrium between the monomeric and condensed states because of their physical properties, either many charged residues (Seq1) or a majority of hydrophobic residues (Seq2), would change the tendency to

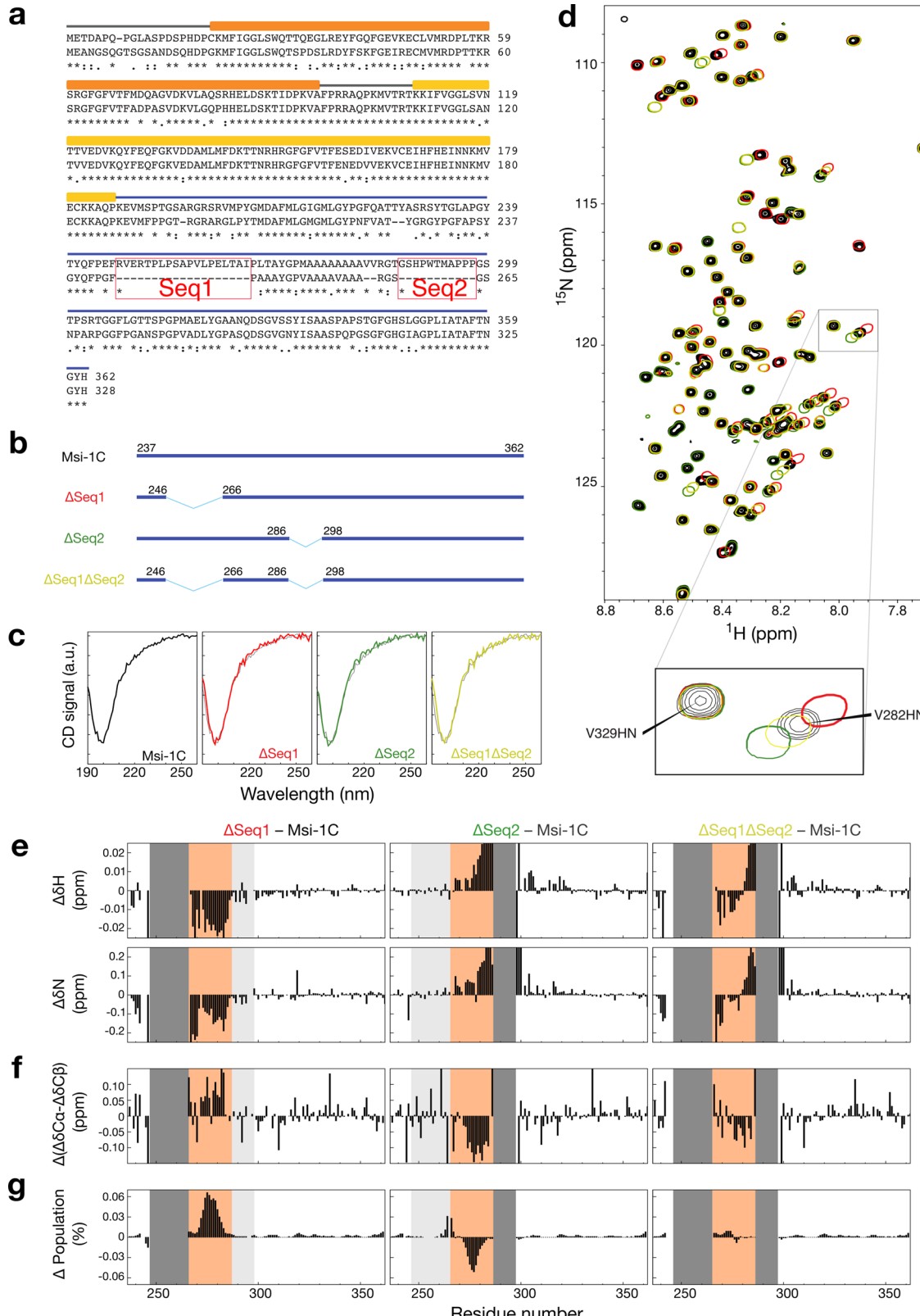

**Fig. 3 The peptides flanking the polyalanine tract tune its α-helical propensity. a** Sequence alignment of the human Musashi proteins. The main difference is the absence of the Seq1 and Seq2 regions in Musashi-2. **b** The designed constructs, with only the preserved residues indicated. **c** Circular dichroism spectra of the different constructs (in black, red, green, and yellow) overlaid on the wild types (in gray). **d** Overlaid NMR HSQC spectra (color-coded as in panel (**b**)), with an expanded view of overlapping (Val-329) and shifted (Val-282) cross-peaks. **e**–**g** Difference between the deletion constructs and the wildtype for **e** proton (ΔδH) and nitrogen (ΔδN) chemical shifts, **f** secondary chemical shift differences between Cα and Cβ; **g** δ2D[34] α-helical propensity scores. The Seq1 and Seq2 regions are indicated in light gray if present, dark gray if missing; the polyalanine region is highlighted in orange. The CD and HSQC spectra were repeated at least three times for all protein samples.

assemble. Although the helicity correlates with self-assembly in some cases[37], the correlation is difficult to identify in our studies. Nevertheless, it is notable that there are no species in which Seq1 or Seq2 are missing alone, hinting that this α-helical propensity may have been fine-tuned.

**The polyalanine region promotes Musashi-1 assembly**. Msi-1C and Msi-2C differ in α-helical propensity and the regions in which they differ most change this propensity, and thus we investigated the effects of removing the polyalanine stretch between Seq1 and Seq2. The CD curve of this variant, denoted ΔSeqA, differs from that of Msi-1C around 220 nm (Fig. 4a), in a similar manner as Msi-2C's does (Fig. 2a), indicating a loss of α-helicity. Except for the truncation sites, the HSQC spectrum overlaps well with Msi-1C's, indicating no further structural change (Fig. 4b, c).

In all the NMR and CD studies, we noticed that the signal intensities did not correlate with protein concentrations. For example, the intensity ratio of HSQC spectra did not match their molar ratio (Supplementary Fig. 5). We thus speculate that higher-order oligomers (NMR-undetectable) are present. We used optical microscopy and indeed observed condensates in each case (Fig. 4d and Supplementary Fig. 6). At pH 5.5, the condensates were all spherical, and fluorescence recovery after photobleaching (FRAP) measurements on proteins labeled with a fluorescent probe (Cy3-NHS) showed that the condensates are dynamic (Fig. 4e). At pH 6.5, on the contrary, the condensates observed for Msi-1C were irregular and showed no fluorescence recovery (Fig. 4d, e), whereas, for Msi-2C and ΔSeq, the condensates remained spherical and dynamic (Fig. 4d and Supplementary Fig. 6). In order to avoid ambiguity in the following discussion, the term "condensates" is used for the dynamic states and "aggregates" for the less dynamic and irreversible state (irregular shapes under the microscope), as suggested[38,39]. We also noticed that the level of fluorescence recovery varied between samples (gray lines in Fig. 4e), indicating that the condensates undergo very rapid sol-gel transitions (Fig. 4e), especially for Msi-1C at pH 5.5 (i.e., quantifying the level of recovery would not be informative). Rapid aggregation has also been reported for other proteins[40–42]. Therefore, to compare the aggregation tendency, we prepared 20 μM samples at pH 5.5 and 6.5 and incubated them at room temperature for different periods. We then centrifuged the samples (at ~12,000 × g, to remove large aggregates but leave small condensates in the supernatant, as confirmed by microscopy) and measured the concentration of the soluble fraction. At pH 5.5, the proteins remained mostly in the supernatant for all incubation times (Fig. 4f, as determined by the absorbance at 280 nm). Condensates in the soluble fraction may affect the absorbance accuracy, thus we have also confirmed the amount of supernatant protein using SDS-PAGE gel (Fig. 4f; uncropped images in Supplementary Fig. 6). At pH 6.5, nearly all Msi-1C molecules aggregated within 1 h of incubation but Msi-2C, and to a lesser extent, the ΔSeqA construct remained partially in the supernatant for longer (at least 24 h; Fig. 4f). These results suggest that the polyalanine region in Msi-1C promotes aggregation. Figure 4g shows an energy landscape representation of the metastable nature of LLPS in these proteins[40]. Their three main states are the monomeric form, the LLPS state (dynamic condensate), and the aggregate form. At pH 5.5, the energy barrier between LLPS and aggregation is high, leaving Msi-1C trapped in the dynamic condensate. At pH 6.5, however, the energy barrier is lower, and Msi-1C aggregates quickly. However, at the same pH, ΔSeqA remains in the dynamic soluble state for longer, indicating that removing the α-helical region restores the energy barrier between

the LLPS and aggregate states. These results suggest that although Msi-1C is less prion-like, its stronger α-helical propensity promotes assembly, regardless of the "price paid" in terms of aggregation[42,43].

## Discussion

**Various properties contribute to functional assembly in IDRs**. Without the constraints of a fixed shape, the IDRs in a paralog family can evolve more freely than structured regions, either gaining new functions or compensating for lost ones. The results obtained here for the Musashi protein family suggest IDRs may have evolved different means of functional assembly. Musashi-1 is found in stress granules, but its IDR has fewer prion-promoting amino acids than many others with this property (e.g., FUS, TDP-43, and hnRNP A1). However, our results suggest that the stronger α-helical propensity of Musashi-1's polyalanine stretch may assist its assembly, as polyalanine tracts are known to contribute to protein self-assembly[44]. On the other hand, Musashi-2's IDR is sufficiently prion-like for assembly despite a lower α-helical propensity (Fig. 4).

Our results also explain a number of biological observations. Musashi proteins are overexpressed for cell renewal and stemness maintenance[22]. Although certain cell types express one or other Musashi paralogs (e.g., Musashi-2 in hematopoietic stem cells[45]), they are functionally redundant when they appear together. For example, Musashi-2 compensates for the proliferation of neural progenitor cells in Musashi-1 double-knockout mice[20]; both Musashi proteins promote colorectal cancer cell growth through the same signaling pathway[46]; complete loss of visual function in photoreceptor cells is only observed in Musashi-1,2 double-knockout mice[47]. The IDRs' different physical properties compensate for the loss of the other, but these differences (e.g., α-helix dominant vs more prion-like) may nevertheless lead to specific interaction mechanisms, for instance, tau protein only interacts with Musashi-1 for transportation into nuclei, whereas tau's interaction with Musashi-1 or −2 leads to different pathological stages in tauopathies[26].

**Polyalanine is commonly involved in promoting self-assembly**. Polyalanine sequences may be a common evolved trait through which RBPs assemble or join membraneless organelles. For example, a sufficiently long polyalanine tract in its IDR enhances the subnuclear targeting properties of RBM4 (RNA-binding motif 4)[48]. Indeed, with glutamine and asparagine, alanine is one of the most frequently repeated amino acids in proteins, and poly-alanine stretches promote self-assembly[44]. Several other amino acids are also helix-promoting, such as methionine, leucine, and glutamine[35]. In the IDR of the extensively studied TDP-43, for example, the "AMMAAAQAALQ" amino acid motif has a strong tendency to form an α-helix, promoting condensation[49–51] regardless of its short polyalanine tract. In order to estimate the frequency of polyalanine appearance, we rapidly searched for this trait in the proteome using a simple scoring function based on amino acid α-helical propensity values derived from our experi-mental observations and consensus studies (Fig. 5a)[35,36]. Using these tentative scores and aggregating for repeats (examples shown in Fig. 5b), we calculated the portion of all IDRs in the human proteome and in RBPs (1542 in total) in which the aggregated polyalanine score reaches 1.5, 2.0, or 2.5 (Fig. 5c). We also analyzed a group of 692 mRNA-binding proteins (mRBPs) because many reported RBPs[42,52,53] with LLPS-related functions reside in this category. Our analysis shows that polyalanine stretches are more common in RBPs, especially mRBPs, than in the general human proteome (Fig. 5c). We also used RaptorX[54] to predict the residues with α-helical propensity among the IDRs of

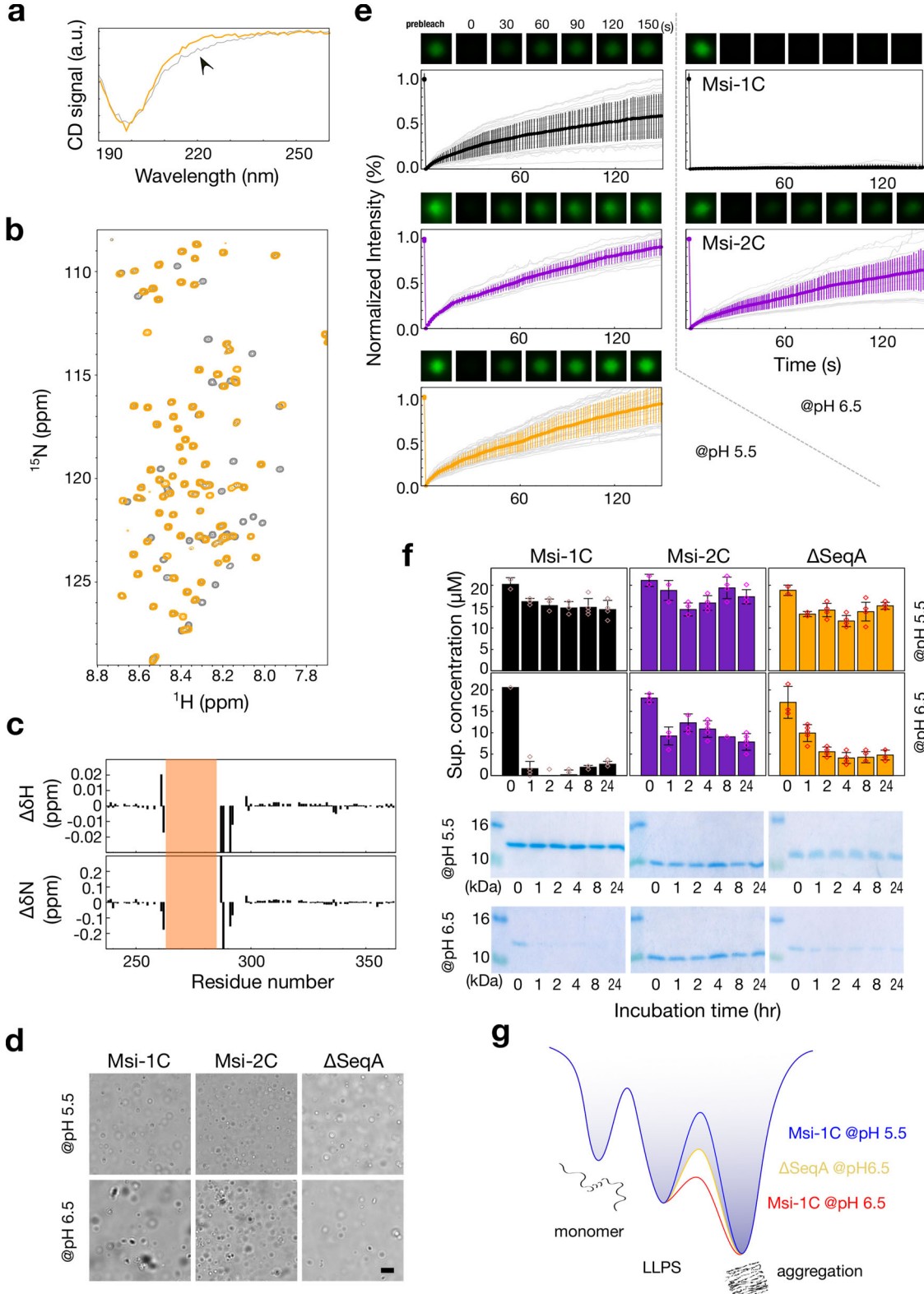

**Fig. 4 The α-helical region promotes self-assembly in Musashi-1's C-terminal domain (Msi-1C).** Overlaid **a** circular dichroism and **b** NMR HSQC spectra of the ΔSeqA construct (orange) and Msi-1C (gray). **c** Chemical shift differences in the proton (ΔδH) and nitrogen (ΔδN) dimensions between ΔSeqA and Msi-1C. The polyalanine region is highlighted in orange. **d** Light micrographs of Msi-1C, Msi-2C, and ΔSeqA at pH 5.5 or 6.5. Scale bar: 10 μm. Experiments were performed at least three times for each protein sample. **e** Fluorescence recovery after photobleaching (FRAP) results in pH 5.5 and 6.5. The colored lines represent the mean ± SD and the gray lines are individual recovery profiles. At least 20 condensates were recorded for each sample. **f** Precipitation assays of Msi-1C, Msi-2C, and ΔSeqA at pH 5.5 and 6.5 after incubation at room temperature for different time periods. Supernatant concentrations were determined after centrifugation (triplicate; mean ± SD) and confirmed by SDS-PAGE. **g** Energy landscape representation of the results.

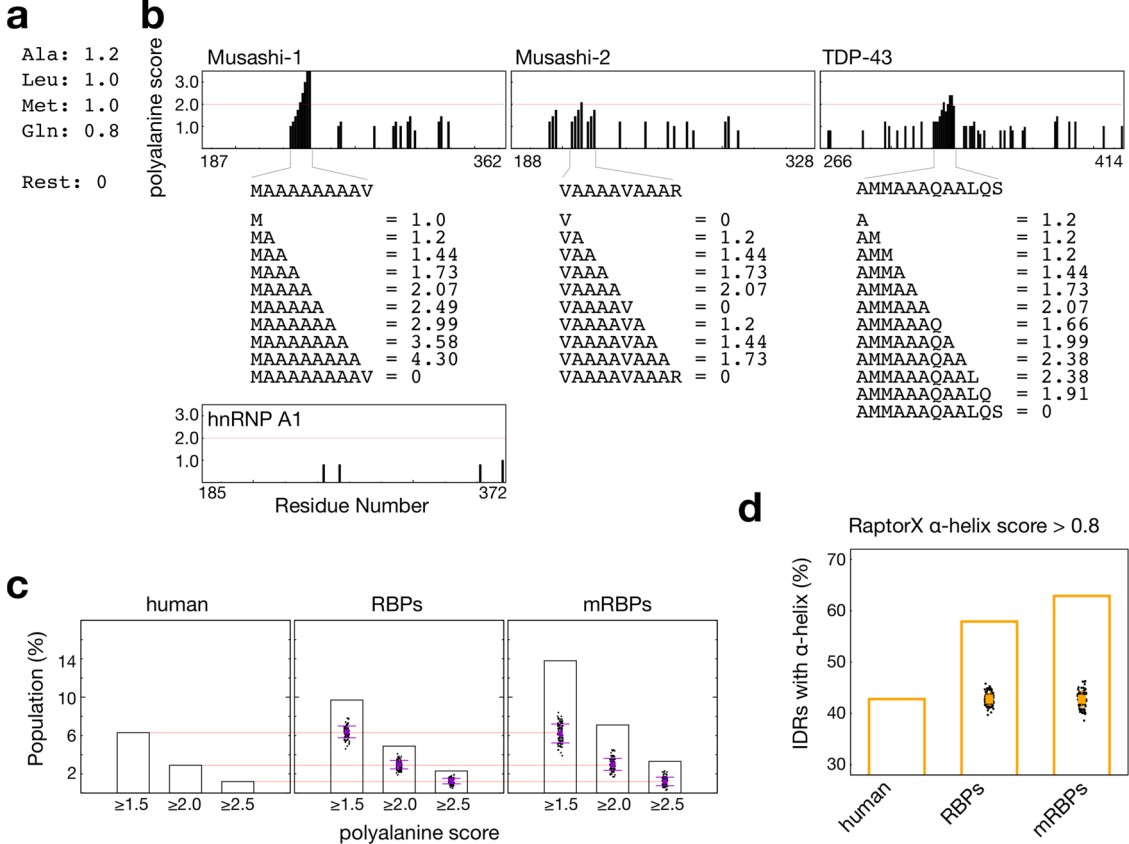

**Fig. 5 Estimated prevalence of polyalanine and α-helix-forming residue stretches in the intrinsically disordered regions. a** Tentative scores are assigned to the amino acids according to the experimental observation to estimate the frequency of polyalanine appearance. **b** Examples of how polyalanine propensity scores were calculated for continuous residue stretches of increasing length in proteins with α-helix-forming regions (Musashi-1, Musashi-2, and TDP-43) or without (hnRNP A1). **c** Prevalence of IDRs with polyalanine scores greater than 2.5, 2.0, and 1.5, in the human proteome, RNA-binding proteins (RBPs), and mRNA-binding proteins (mRBPs). **d** Population of IDRs with α-helical elements longer than five consecutive residues in the human proteome, RBPs, and mRBPs. The point clouds, square points, and error bars in the RBP and mRBP panels are the distribution, mean, and standard deviation of the polyalanine (panel **c**) and α-helix scores (panel **d**) of 100 random selections of sets of 1542 or 689 proteins (the RBP and mRBP sample sizes) from the human proteome (as negative controls).

the human proteome, RBPs, and mRBPs. We set arbitrary criteria (all other criteria show the same trend) to count those IDRs with at least five consecutive residues predicted higher than 0.8 (1.0 is the full scale) α-helical propensity in the program. Figure 5d shows the same tendency as the polyalanine score: mRBPs' or RBPs' IDRs are more likely to have α-helical elements within them. In these analyses, the difference is higher than the deviation of 100 random selections of 1542 or 689 proteins from the human proteome (the sample sizes considered for the RBPs and mRBPs, respectively; Fig. 5c, d). We attribute the results to that RBPs often join biomolecular condensates for their functions[55], and polyalanine/α-helix is one feature of IDRs that contributes to self-assembly.

**When did Musashi IDRs diverge?** The IDRs of Musashi proteins may have respectively gained α-helicity or become less prion-like. Which came first? The answer is perhaps hidden in the primary sequence. Using the RRM from *D. melanogaster* Musashi once again, we searched for orthologs in the phylum Chordata, a higher taxonomic rank than the subphylum of vertebrates in which the Musashi paralogs arise (Fig. 1a). *Amphioxus* (the lancelet), a model organism for primitive chordates, has a Musashi homolog which does not have a long polyalanine tract but is prion-like, similar to that of the nematode and fruit fly (Fig. 6a, b and Supplementary Fig. 7). This suggests that the Musashi

paralogs may have arisen after the appearance of chordates. In ghost sharks indeed, primitive vertebrates with the two Musashi paralogs, Musashi-1 has a polyalanine tract that is not prion-like (Fig. 6c and Supplementary Fig. 7), whereas interestingly, Musashi-2 has a polyalanine tract and is prion-like. Collectively, these results suggest that the polyalanine stretch may have appeared in the paralogs alongside primitive prion propensity before Musashi-l lost its prion-promoting amino acids, whereas a disruption in the α-helical region reduced Musashi-2's tendency to self-assemble.

**"Distant relatives" in the Musashi family.** According to the RBP census study[11], many other RBPs were grouped in the paralog family of Musashi proteins based on sequence similarity. We confirmed this by searching for sequences similar to Musashi-1's in the human proteome. Other than Musashi-2 (71.3% identity), the top hits are DAZ-associated protein 1 (DAZAP-1, 42.6%), several heterogeneous nuclear ribonucleoproteins (hnRNP A0, A1, A2…; 39.0–42.5%) and TDP-43 (34.9%) (Fig. 6d), agreeing with the previous study[11]. Accordingly, we compared the IDRs' properties of these Musashi protein's "distant relatives": Studies of the low complexity IDRs of hnRNP A1 and A2 are pioneering examples in the emerging field of protein LLPS[41,52,56]. They are also a good model for studying LLPS theory, such as the effects of the number and distribution of aromatic residues[57]. TDP-43's

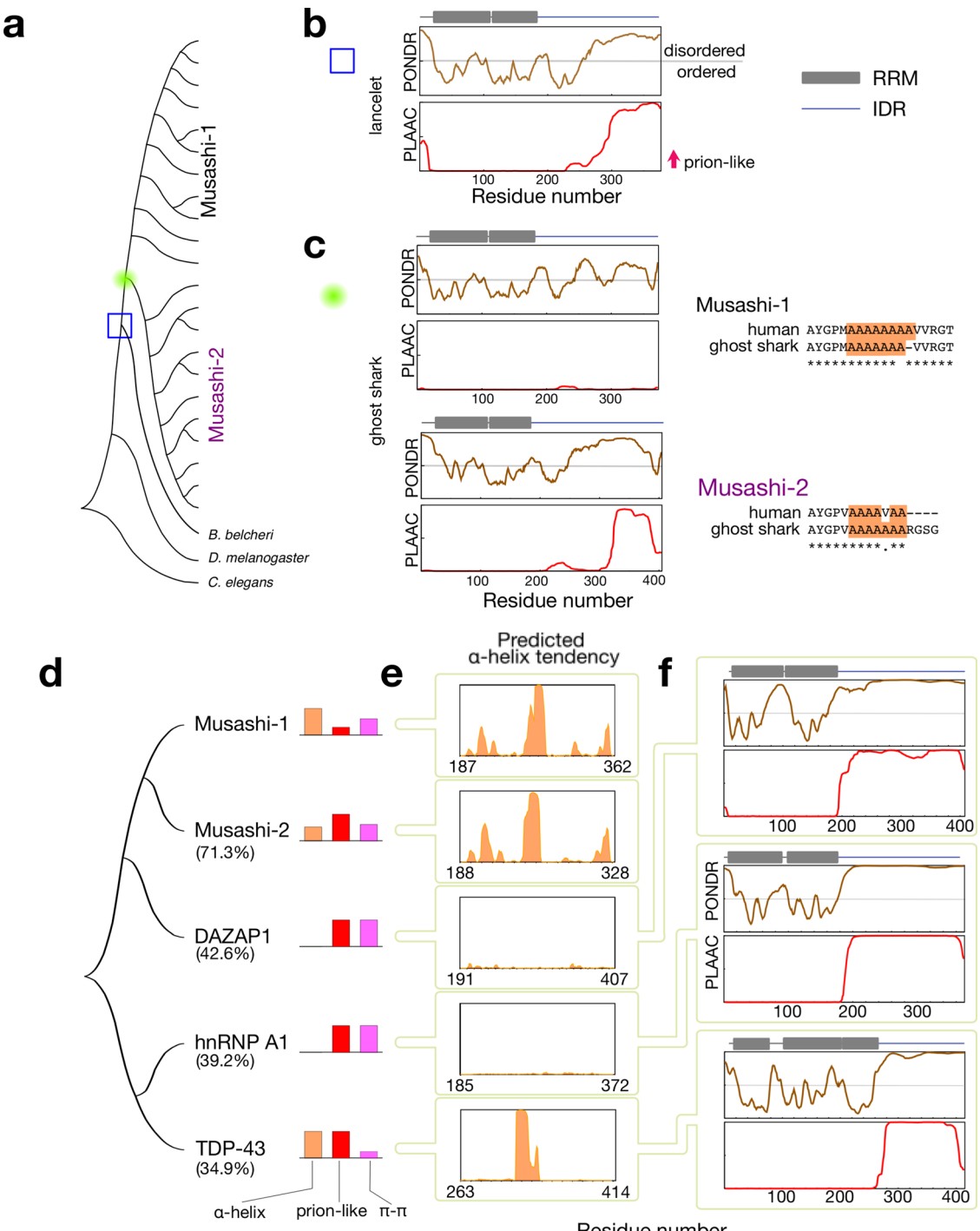

**Fig. 6 Sequence analysis of the Musashi family in primitive model organisms and distantly related paralogs. a** Phylogenetic tree of the Musashi family as shown in Fig. 1a with the additional lineages analyzed: a primitive chordate (the lancelet, blue box, panel (**b**)) and a primitive vertebrate (the ghost shark, green circle, panel (**c**)). **b, c** PONDR sequence disorder and PLAAC prion-likeness scores of **b** the lancelet (*Buccinum belcheri*) and **c** the ghost shark (*Callorhinchus milii*). The RNA recognition motifs (based on PROSITE analysis[73]) are indicated by gray squares. The alignment of the polyalanine regions (highlighted in orange) of human and ghost shark Musashi proteins are also shown. **d** Phylogenetic tree of Musashi paralogs with high sequence similarity found in the human proteome. The percentages shown are the levels of sequence similarity. The importance of different drivers of assembly in their IDRs is indicated by colored bars representing their α-helical propensity (orange), prion-likeness (red), and the number of aromatic residues (purple). **e** α-helical propensity predicted using RaptorX[54]. **f** Sequence disorder and prion-likeness for DAZAP-1, hnRNP A1, and TDP-43 (c.f. Fig. 1b for Musashi-1 and −2).

IDR is another extensively studied example. TDP-43's IDR differs from hnRNP A1 or A2's in that, as well as being prion-like, it encompasses a short α-helical region that contributes to self-association and LLPS[49,51]. Furthermore, TDP-43's LLPS is mediated by just a few aromatic residues[50], a feature we have attributed to the presence of the α-helix, which promotes intermolecular contacts[50] (Fig. 6d). The present study adds the biophysical properties of the two Musashi proteins to existing information on this distantly related protein family. Although DAZPA1 has been far less studied than other members, its

C-terminal domain is critical to its function in interacting with eIF4G[58] and potentially interacts with many RBPs, including hnRNP A1[59]. Moreover, bioinformatic analysis shows that this protein's C-terminal domain is disordered and prion-like, with many aromatic residues but no predicted α-helix, similar to that of hnRNP A1 (Fig. 6e, f and Supplementary Fig. 8). Although the IDRs of these RBPs have different properties, they undergo LLPS. As a conclusion, we suggest that IDRs evolve whatever traits are beneficial to function, which accords with François Jacob's statement that "evolution does not produce novelties from scratch"[60]. He was referring to the diversity of all lifeforms but his words also apply to IDRs, which acquire new functions through evolutionary "tinkering"[60] between prion-likeness, α-helicity, aromatic residues, etc. Our work could be used as a template to investigate IDRs in other paralogs and how their functions have diversified or been preserved during evolution.

## Methods

**Bioinformatics analysis**. The primary sequences were obtained from UniProt with the associated entries listed in Supplementary Table 1. Levels of structural disorder and prion-likeness were respectively analyzed with the PONDR[61] and PLAAC[30] webservers. The phylogenic trees were construed using the neighbor-joining method in MEGA X[62].

In analyzing the polyalanine and α-helical propensities, all human protein sequences were retrieved from UniProt (UniProtKB_2021_01, download date: 2021/02/06, 20396 sequences in total) and separated into disordered and ordered regions based on PONDR predictions (VSL2)[61]. The predicted disordered regions longer than 40 consecutive residues were analyzed. The tentative polyalanine scoring (Fig. 5a) of the sequences was done using in-house scripts to estimate the frequency of polyalanine appearance. The α-helical propensities were predicted using RaptorX[54]. An IDR in a protein having at least five residues in a raw predicted as α-helix (with a value higher than 0.8 out of 1.0) is counted in our analysis.

**DNA constructs**. The Msi-1C construct in this study (residues number 237 to 362) was cloned from a previous longer construct (194–362)[63]. We removed a strongly hydrophobic region reported to contain binding sites for PABP and GLD2[22], which is irrelevant to the present study and whose removal improves the protein's solubility. The ΔSeq1, ΔSeq2, ΔSeq1ΔSeq2, and ΔSeqA constructs were prepared with designed primers (Supplementary Table 3). Msi-2 cDNA (residues 234–328, according to the alignment with Msi-1C) was purchased from OriGene. All these variants were constructed in a pET21 vector backbone with a hexahistidine tag on the C-terminus of the expressed protein. All constructs were fully sequenced.

**Protein expression and purification**. All constructs were purified using the same protocol, which has been described in detail elsewhere[63]. In short, transformed *E. coli* BL21(DE3) cells were grown at 37 °C until the OD reached 0.6 and were induced with a final concentration of 1 mM isopropyl-β-D-1-thiogalactopyranoside at 25 °C overnight. The cells were harvested and lysed by sonication. After centrifugation, the inclusion bodies were dissolved using 20 mM Tris buffer at pH 8 with 8 M urea (buffer A). After a second centrifugation, the supernatant was filtered with a 0.45 μm filter and loaded onto a nickel-charged immobilized metal affinity chromatography column (Qiagen). After washing with ten column volumes of buffer A, the samples were eluted with 5 column volumes of buffer B (buffer A with 500 mM imidazole). The eluted samples were acidified with trifluoroacetic acid (down to pH ~3), loaded onto a C4 reverse-phase column (Thermo Scientific Inc.), and eluted with a gradient of acetonitrile (from 0 to 100%, mixed with triple-distilled water) by HPLC and then lyophilized. The lyophilized samples were stored in a dry cabinet until use. For all experiments, powder samples were dissolved in 20 mM MES-NaOH at pH 5.5 or 6.5. The protein concentration was determined from the absorbance at 280 nm measured with a Nanodrop spectrometer (Thermo Scientific Inc.).

**Circular dichroism spectroscopy**. Circular dichroism spectra were recorded using an AVIV model 410 spectropolarimeter with a 0.1 mm cuvette. Data were collected between 190 and 260 nm with an interval of 1 nm. Ten measurements were co-added for each data point. All spectra were recorded at 283 K and the samples were kept in a water bath at 283 K between measurements. All experiments were performed in triplicate.

**NMR spectroscopy, chemical shift assignment, and data analysis**. $^{15}N$-edited HSQC spectra were recorded using the standard pulse sequence with WATER-GATE solvent suppression[64,65]. Chemical shifts were assigned using standard HNCA, HN(CO)CA, HNCO, HN(CA)CO, CBCA(CO)NH, and HNCACB experiments acquired with non-uniform sampling (25%)[66,67]. All data were recorded using a Bruker AVIII 600 MHz spectrometer with a cryogenic probe.

The data were processed using NMRPipe[68]. Chemical shifts were assigned using the automated assignment scheme[69] implemented in NMRFAM-Sparky[70], and then confirmed manually. Secondary chemical shift analysis was performed using Kjaergaard et al.'s database of random-coil shifts[71]. Secondary structure populations were estimated using δ2D[34]. No chemical shifts were missing around the critical α-helical region, such that the secondary structure estimates for the constructs were made using the same number of chemical shifts.

**Microscopy**. Protein samples were loaded onto mPEG-passivated slides[72]. Micrographs were collected using an Olympus BX51 microscope with a 40× long-working-distance objective lens.

**Microscopy and fluorescence recovery after photobleaching (FRAP) experiments**. For the fluorescent dye labeling, 2.5 mg lyophilized protein was dissolved in 0.1 M sodium phosphate with 6 M guanidine hydrochloride at pH 8.3 and mixed with 15.6 mM (~1 mg in total) of Cy3-NHS (Lumiprobe) overnight at room temperature. The excess Cy3-NHS was removed and the buffer was exchanged with a 20 mM MES buffer at pH 5.5 using a PD-10 column (GE Healthcare). The typical labeling efficiency, determined from the ratio of extinction coefficients measured for the protein and the fluorescence dye (Cy3: 150,000 $M^{-1}$ $cm^{-1}$ at 550 nm) was ~20%. The Cy3-labeled sample was aliquoted, flash-frozen, and stored at −80 °C before usage.

The lyophilized samples were dissolved in 20 mM MES buffer at pH 5.5 or pH 6.5 and mixed with a Cy3-labeled sample with a final concentration of 20 μM and ~1% Cy3-labeled protein. The samples were then loaded onto ultraclean coverslips and observed with an *iLas* multi-modal of total internal reflection fluorescence (Roper Scientific, Inc.)/spinning disk (CSUX1, Yokogawa) confocal microscope (Ti-E, Nikon) equipped with 100 × 1.49NA plan objective lens (Nikon). The condensates were bleached with a 561 nm laser. Images were acquired at one-second intervals using an Evolve EMCCD camera (Photometrics) and were analyzed using the Metamorph software (Molecular Devices, LLC).

**Aggregation assays**. Lyophilized samples were dissolved in 20 mM MES buffer at pH 5.5 or pH 6.5 and stored at room temperature for different times. The samples were then centrifuged at 12,000 × *g* for 5 min. Supernatant concentrations were determined using a NanoDrop spectrometer. Supernatant samples were also analyzed using SDS-PAGE gels to confirm the amount of protein present. All experiments were performed in triplicate.

**Statistics and reproducibility**. All NMR HSQC spectra, CD experiments, and microscope observations were repeated at least three times. The reproducibility of these types of biophysical experiments is high. The FRAP experiments were repeated from three independently prepared samples. At least ten condensates were recorded to obtain the mean ± standard deviation recovery curves. Precipitation assays were performed at least three times from independently prepared samples. The data were given as the mean ± standard deviation. The individual data points were also shown to present data distribution.

**Reporting summary**. Further information on research design is available in the Nature Research Reporting Summary linked to this article.

## Data availability

The chemical shift assignments for generating Figs. 2b, 2c, 2d, 3d, 3e, 3f, 3g are deposited in the Biological Magnetic Resonance Bank (BMRB): 51204, 51205, 51206, 51207, 51208. Scripts for reproducing Fig. 5 are deposited in http://github.com/allmwh/helix_score.

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

## Acknowledgements
The authors thank Prof. Won-Jing Wang (NYCU) for access to the microscope, Academia Sinica High-Field NMR Center for technical support (HFNMRC is funded by Academia Sinica Core Facility and Innovative Instrument Project (AS-CFII-108-112)), Chen-Yu Hung for microscope setting, and Dr. Tsai-Chen Chen for initial work on this project. This work was supported by the Ministry of Science and Technology of Taiwan (109-2113-M-010-003 and 110-2113-M-A49A-504-MY3 to J.-r.H. and 109-2326-B-010-002 -MY3 to J.-C.K.).

## Author contributions
J.-r.H. conceived the project and wrote the manuscript. S.-H.C., W.-L.H., Y.-C.S., J.-C.K., and J.-r.H. collected and analyzed the data.

## Competing interests
The authors declare no competing interests.
