## [Peer Review File · Communications Biology]

Reviewers' comments:

Reviewer #1 (Remarks to the Author):

The authors have conducted a good and thoughtful study which I will attempt to summarize as follows. The phenomenon of liquid-liquid phase separation has recently been recognized as a major contributor to cellular organization through the formation of membraneless organelles. Many of these membraneless organelles involve RNA-binding proteins (RBPs), either as major drivers of phase separation, or as "clients" which partition into membraneless organelles, and which are the focus of this study. These RBPs commonly contain an intrinsically disordered region that contributes to phase separation propensity.

The intrinsic disorder of a protein region allows not only for nonspecific interactions involved in phase separation, but also makes them more prone to evolution. This study incorporates both of these aspects and presents a hypothesis regarding the evolution of IDRs for certain functions, including phase separation. Specifically, in the case of the Musashi protein family, they study how two copies of a gene may diverge in sequence, but give rise to proteins with similar functions. This is a relatively new line of thinking in the field, and would be well-suited for Communications Biology. The conclusions are well-reasoned in my opinion, but I have some questions regarding the techniques applied. I also believe the authors could make some changes to improve the communication of the results and conclusions.

I have listed my concerns and comments below for the authors to address:

Major:

1. The first couple sentences in the introduction are confusing and should be rewritten to avoid jargon and to be more concise.

1a. Something like "Recurrent gene duplications over many generations create paralogs in a genome, which can give rise to a myriad of new protein functions." may work better for the first sentence.

1b. What is a pseudogene?

1c. Why have IDR paralogs not been explored before?

2. Paragraph 2 of introduction: What is the difference between the 22% being in a paralog family, and 48% being a paralog?

3. The introduction and first results section could benefit from a brief description of what "prion-likeness" is, and how it relates to phase separation.

4. The conclusion from the results subsection "The non-conserved regions flanking the polyalanine tracts tune their α -helical propensity" is very interesting. I'm curious that the presence of both flanking stretches does not impart additional helical stabilization, and thus no net change. Could this have some effect of stabilizing against conditions that may promote or weaken helix formation? The process of phase separation itself can promote or destabilize helicity [Conicella et al. PNAS, 2020]. Could the authors comment on why this might be?

5. In the results subsection "The polyalanine region promotes Musashi-1 assembly", the authors state that they "observed condensates in each case" Were all NMR experiments done with a system containing condensates? Some recent work has shown that small differences can arise in NMR spectra due to phase separation [Murthy et al. NSMB, 2019]. Could the authors show that the presence of condensates in the NMR and CD spectra does not cause a significant difference? This is especially significant since α -helical content can be promoted by phase separation as seen with TDP-43 [Conicella et al. PNAS, 2020].

6. In the centrifugation experiments, is the saturation concentration what is being measured in some of the cases? The text states that centrifugation "remove[s] large aggregates but leave[s] small condensates in the supernatant" When measuring the protein concentration, will the presence of small condensates affect the measurement of protein concentration? Later on in the same paragraph, it states that "At pH 5.5, the proteins remained mostly soluble". Does this mean that there were no aggregates or condensates present? Some effort should be made to clarify this section.

7. A distinction should be made between gelation and aggregation in the last subsection of results. Is centrifugation removing aggregates or gel-like condensates?

Minor:

1. I would disagree that "little is known about the roles of intrinsically disordered regions (IDRs)" as stated in the abstract. Much of the recent research that is cited in this manuscript would go against that. There is still much to learn would be a more accurate phrasing.

2. Consider rearranging the abstract to discuss IDRs generally before discussing the specific Musashi family. Thus, I would move the last 2 sentences: "Without a folded structure... such as liquid-liquid phase separation." to before the sentence that introduces the Musashi family.

3. Paragraph 3 of introduction: If I understand correctly, two-member families is referring to a protein family which has been duplicated once, and thus has two instances in the genome. It is not completely clear from the wording that it means this however. Perhaps rephrase it as "families of proteins that have two members" or elaborating further on what is specifically being considered here.

4. In Fig. 1c, the colors of the pie charts are hard to distinguish between. It may be best to include the letters for each chart as in the one on the bottom-left.

5. Fig 2a doesn't need two panels since both CD spectra are shown on the second one.

6. In the second subsection of results, it mentions a β -sheet promoting valine. Is there any significance to this other than helix-breaking? Could it contribute to aggregation and formation of amyloids perhaps? This is not discussed further in the manuscript.

7. Rephrase "residue stretches" in the first sentence of the third subsection of results.

8. Consider moving the FRAP data from Supp. Fig. 6 to the main text since it seems to be a key result that Musashi-1 with the helical region forms aggregates or dynamically-arrested condensates.

Reviewer #2 (Remarks to the Author):

This article analyzed the IDR regions of Musashi protein family from the aspect of molecular biology, molecular evolution, and bioinformatic. The results are interesting and can contribute the elucidations of functions of IDRs. Although I am positive about accepting the article, I would like the authors to consider the following points.

1) I cannot agree the section of "a-helicity is commonly involved in promoting self-assembly" in p8. It is unclear what the helical score, however, it seems multiplying a helical propensity show in Fig.5a of a site in interest and that of the previous residue. You can easily find the helix propensity by using the AAindex database (<https://www.genome.jp/aaindex/>, search with helix or something), and that of A and E of the classical parameter of "Chou and Fasman" are 1.42 and 1.51, and the other parameters recently published show similar. Then, the parameters shown in Fig5.a cannot be accepted. Since the state-of-the art secondary structure prediction programs such as psi-pred and others have accuracy in the practical use, I recommend using one of these programs to conduct the proteome wide analysis.

2) The clouds of Fig5c are not visible, and what is mRBP. I cannot find this term before P8 and no explanation is found.

3) It is better to have organism labels in the alignments in SupFig.2.

Reviewer #3 (Remarks to the Author):

Comment:

Shih-Hui Chiu and co-authors in their current work have focused on how IDRs evolve in protein paralogs and how the structured segments of the protein buffer the variational properties inflicted by the disordered segments in the protein. This is a very interesting and relevant question in the context of the cellular unfoldome and how it evolves. The authors in their test case have worked on Musashi-1 paralogs and have shown lower prion-like tendency of Musashi-1's IDRs compared to that in Musashi-2's and it is compensated by its higher alpha-helical propensity to assist their assembly. The problem/question is introduced with an interesting and clinically relevant example. This puts the point that protein paralogs in general have been studied but not in the context of disordered systems (IDP).

Available literature talks about the paralog proteins in general. This paper talks about the contribution of IDRs in modulating the behavior of the paralogs and how it impacts physico-chemical properties across the scales including condensate forming tendency. This is an interesting read from the perspectives of the physics of the disordered systems. The article is well composed and written in a way so that the message can reach out to more general readership. The article can be considered for publication but there are certain things which needs to be addressed.

Points and Suggestions:

1. Although the authors have explained the biological features associated with Masashi protein family, it would be better if they can put this in the perspective of other proteins. In the discussion section the authors have compared Masashi family traits with other proteins, but it would be better to put some of this information in the introduction section as well. This would make things clearer for a broader readership and to disordered systems physics community.
2. RNA molecules inherently have their own secondary structures. To interact RNA binding proteins sometimes need to utilize their less structured segment to have conformational complementarity. It would be great if this relation is discussed in some of the segments.
3. Although authors in the discussion have talked about other proteins, especially in the context of liquid-liquid-phase separation and condensate forming tendency, a more detailed comparisons with some other model protein as parallel case study would have been interesting and would have strengthened the concept.
4. In the abstract authors have used the phrase "Without folded structure to restrain": Does that mean how much of the protein is folded? It would be interesting to look at the type of fold or super-secondary structures inside the folded segment are mostly evolutionarily selected for these kinds of proteins.
5. It would be interesting if the sequences are scanned for slow codons or kinetic traps and the paralogs are analyzed om in the context of kinetics of folding.
6. Did the authors look for residue positions in the sequence space in the context of stabilizing/destabilizing mutations by deploying some deep mutational scans? That would be interesting in the context of IDRs as inherently they should have higher mutational tolerance owing to low complexity in structures. But on the other hand, their functional role would not allow for few select set of mutational events. A deep mutational analysis would be interesting in this context.
Further I recommend carrying out co-variational analysis to capture the dynamics of sequence variations in the IDR regions and how that correlate to pair-wise linked positions in the protein structure.
7. While comparing Musashi-1 with other proteins in the context of LLPS authors must make things clearer. This section in the discussion forms an important component of the paper and contains some key messages.

8. It would be interesting if authors touch upon the sequence variations in the IDR segments of the paralogs in the context of protein quality control.

Reviewers' comments:

Reviewer #1 (Remarks to the Author):

The authors have conducted a good and thoughtful study which I will attempt to summarize as follows. The phenomenon of liquid-liquid phase separation has recently been recognized as a major contributor to cellular organization through the formation of membraneless organelles. Many of these membraneless organelles involve RNA-binding proteins (RBPs), either as major drivers of phase separation, or as "clients" which partition into membraneless organelles, and which are the focus of this study. These RBPs commonly contain an intrinsically disordered region that contributes to phase separation propensity.

The intrinsic disorder of a protein region allows not only for nonspecific interactions involved in phase separation, but also makes them more prone to evolution. This study incorporates both of these aspects and presents a hypothesis regarding the evolution of IDRs for certain functions, including phase separation. Specifically, in the case of the Musashi protein family, they study how two copies of a gene may diverge in sequence, but give rise to proteins with similar functions. This is a relatively new line of thinking in the field, and would be well-suited for Communications Biology. The conclusions are well-reasoned in my opinion, but I have some questions regarding the techniques applied. I also believe the authors could make some changes to improve the communication of the results and conclusions. I have listed my concerns and comments below for the authors to address:

We thank the reviewer for this positive and clear summary.

Major:

1. The first couple sentences in the introduction are confusing and should be rewritten to avoid jargon and to be more concise.
 - 1a. Something like "Recurrent gene duplications over many generations create paralogs in a genome, which can give rise to a myriad of new protein functions." may work better for the first sentence.
 - 1b. What is a pseudogene?

We have rephrased the article's opening as the reviewer suggested and have added a short description to define "pseudogene" for more general readers. Furthermore, we have rewritten some sentences (highlighted in red) to make them more readable. (e.g. To start with a general term: we have changed "The paralog a- and b-globin chains that form the hemoglobin tetramer are a textbook example" to "The hemoglobin, composing paralog a- and b-globin chains, is a textbook example....."; To put unnecessary gene name to parenthesis only for reference: we have changed "....., but not the b-thalassemia, which results from the loss of b-globin encoding in only *HBB*" to "....., but not the b-thalassemia, which results from the loss of the only one b-globin encoding gene (*HBB*)").

1c. Why have IDR paralogs not been explored before?

Intrinsically disordered regions are relatively less studied than their folded counterparts (<https://www.nature.com/articles/471151a>). In this relatively new field, our group starts this new line in looking at their paralogs. We think this is the novelty of our article. To avoid extensive explaining why IDR paralogs have not been explored, we have revised the last sentence in the introduction as “....., **remain largely ignored.**” to address this point.

2. Paragraph 2 of introduction: What is the difference between the 22% being in a paralog family, and 48% being a paralog?

We surveyed the census study of human RBPs (reference 11) and found that 341 out of the total 1542 RBPs (22%) are annotated in the same paralog families (OrthoMCL database). If only based on sequence similarity (the same result as in the reference 11), 749 out of 1542 (48%) are paralogs. To clarify this difference, we have rephrased the sentences as: “**In the manually curated list of RBPs (1,542 RBPs in total), more than 22 % (341/1542) are annotated in one of the paralog families in the OrthoMCL database, and more than 48 % are paralogs (749/1542) as defined in the original census study based on sequence similarity (see Supplementary Data Sheets S1 and S2 for the lists).**”

3. The introduction and first results section could benefit from a brief description of what "prion-likeness" is, and how it relates to phase separation.

We have added a short phrase in the introduction “**(the level of amino-acid composition resembling that of prion proteins)**” and have revised the sentence as “.....**The stress-granule-related proteins often possess a prion-like domain, which is similar to prion behavior to assist assembly but does not necessarily aggregate.**” in the result section with related citations (refs. 28, 29).

4. The conclusion from the results subsection "The non-conserved regions flanking the polyalanine tracts tune their α -helical propensity" is very interesting. I'm curious that the presence of both flanking stretches does not impart additional helical stabilization, and thus no net change. Could this have some effect of stabilizing against conditions that may promote or weaken helix formation? The process of phase separation itself can promote or destabilize helicity [Conicella et al. PNAS, 2020]. Could the authors comment on why this might be?

The segments (Seq1 and Seq2) have different physical properties. The higher tendency to phase separation may promote its helicity and vice versa. A possible scenario is that the removal of Seq1 (many charged residues)

enhances the self-assembly (because of less repulsive force) and increases helicity, whereas deleting Seq2 (predominantly hydrophobic) enhance the solubility and thus less helicity. However, because of the technical limitation of detecting the helicity in the condensates (see comment 5 below), we think it would be over-interpreting the results on the level of helicity and LLPS.

Nevertheless, we have added some sentences to address this issue:

“.....Both these regions modify the a-helical propensity probably by altering the equilibrium between the monomeric and condensed states because their physical properties, either many charged residues (Seq1) or a majority of hydrophobic residues (Seq2), would change the tendency to assemble.

Although the helicity correlates with self-assembly in some cases (Conicella et al. 2020), the correlation is difficult to identify in our studies. Nevertheless, it is notable that there are no species in which Seq1 or Seq2 are missing alone, hinting that this a-helical propensity may have been fine-tuned.”

5. In the results subsection "The polyalanine region promotes Musashi-1 assembly", the authors state that they "observed condensates in each case" Were all NMR experiments done with a system containing condensates? Some recent work has shown that small differences can arise in NMR spectra due to phase separation [Murthy et al. NSMB, 2019]. Could the authors show that the presence of condensates in the NMR and CD spectra does not cause a significant difference? This is especially significant since α -helical content can be promoted by phase separation as seen with TDP-43 [Conicella et al. PNAS, 2020].

According to our microscope study, the NMR sample (20 or 100 μ M) should contain condensate. There may be a concentration threshold that all proteins are monomeric in solution. However, below 20 μ M, it is difficult for the NMR studies. We have compared the HSQC spectra between 20 and 100 μ M samples, and no difference is observed (see the new Supplementary Fig. 5). However, the intensity ratio varies among different samples. This is probably expected because what we detected is the “monomeric” samples in solution. Those molecules within condensates cannot be detected by NMR because they have a very fast transverse relaxation rate (due to the large size), i.e. the signal belonging to the condensates are not detectable.

Our system differs from [Murthy et al. NSMB, 2019]. In their study, the condensates were centrifuged and collected. Therefore, although the signal is weak due to the fast relaxation, their samples' extremely high concentration (>200 mg/ml; ~16 mM) overcome this limit. However, we cannot reach such a high concentration. Musashi proteins aggregate quickly when the concentration is increased. Therefore, we cannot observe what the protein

“looks like” in the condensates and obtain the helical information. This limitation is also applied to CD. The accurate concentration measured in the CD spectrum is also affected by the number of condensates present. Therefore, we cannot accurately quantify the secondary structure levels (for example, using the BeStSel webserver). Accordingly, in this study, we only qualitatively compared the overall shape of CD curves but not the secondary structural levels. We have addressed this limitation “In all the NMR and CD studies, we noticed that the signal intensities did not correlate with protein concentrations. For example, the intensity ratio of HSQC spectra did not match their molar ratio (Supplementary Fig. 5). We thus speculate that higher-order oligomers (NMR-undetectable) are present.” and have added the comparison of high/low concentrations of NMR spectra and their intensity ratio in the revised Supplementary Fig. 5.

6. In the centrifugation experiments, is the saturation concentration what is being measured in some of the cases? The text states that centrifugation “remove[s] large aggregates but leave[s] small condensates in the supernatant” When measuring the protein concentration, will the presence of small condensates affect the measurement of protein concentration? Later on in the same paragraph, it states that “At pH 5.5, the proteins remained mostly soluble”. Does this mean that there were no aggregates or condensates present? Some effort should be made to clarify this section.

7. A distinction should be made between gelation and aggregation in the last subsection of results. Is centrifugation removing aggregates or gel-like condensates?

Comments 6 and 7 are related; we reply together here. We thank the reviewer for pointing this out. Our terms were ambiguous. We used the “aggregates” for those irreversible and with irregular shapes under the microscope but used “soluble” for those condensates in the supernatant. The aggregates can be removed by centrifugation but not the condensates. We have now changed “soluble” to “in the supernatant”. In the revised version, we have also added, “In order to avoid ambiguity in the following discussion, the term “condensates” is used for the dynamic states and “aggregates” for the less dynamic and irreversible state (irregular shapes under the microscope), as suggested (ref 38,39).” to define the terms.

We agree that the condensates in the supernatant affect the concentration determined using absorbance at 280 nm. Therefore, in our study, we also used SDS-PAGE to confirm the samples in the supernatant. The SDS-PAGE assay was in the Supplementary Materials, but we have moved them to the revised Figure 4 to resolve this concern.

Minor:

1. I would disagree that "little is known about the roles of intrinsically disordered regions (IDRs)" as stated in the abstract. Much of the recent research that is cited in this manuscript would go against that. There is still much to learn would be a more accurate phrasing.

We thank reviewer for pointing this sentence out. We mean "the roles of IDRs play in the protein paralogs", instead of saying that IDRs are less studied. We have rephrased this sentence to clarify this point ".....but little is known about the roles that intrinsically disordered regions (IDRs) play in their paralogs."

2. Consider rearranging the abstract to discuss IDRs generally before discussing the specific Musashi family. Thus, I would move the last 2 sentences: "Without a folded structure... such as liquid-liquid phase separation." to before the sentence that introduces the Musashi family.

Based on the reviewer's suggestion, we have moved the sentence starting with "Without..." before introducing Musashi and adding a sentence about our challenge: "Without a folded structure to restrain them, IDRs mutate more diversely along with evolution. However, how the diversity of IDRs in a paralog family affects their functions is unexplored." The final sentence is our conclusion, and thus we still keep it at the end. We have revised it as "Our work suggests that, no matter how diverse they become, IDRs could evolve different traits to a converged function, such as liquid-liquid phase separation."

3. Paragraph 3 of introduction: If I understand correctly, two-member families is referring to a protein family which has been duplicated once, and thus has two instances in the genome. It is not completely clear from the wording that it means this however. Perhaps rephrase it as "families of proteins that have two members" or elaborating further on what is specifically being considered here.

We have changed this phrase accordingly.

4. In Fig. 1c, the colors of the pie charts are hard to distinguish between. It may be best to include the letters for each chart as in the one on the bottom-left.

Adding all letters in this panel would make this figure crowded. Fig. 1c aims to show a similar color pattern for the RRM of Musashi 1 and 2 (left column) and their roughly even distribution. In contrast, their IDRs (right column) show more different amino-acid populations. We only highlighted Q and N because they are most important for the prion-like tendency. Therefore, we think adding letters for each chart is not necessary.

5. Fig 2a doesn't need two panels since both CD spectra are shown on the second one.

We agree with this suggestion, but we think the layout (aligned above their correspondent assigned NMR spectra in Fig 2b) is better.

6. In the second subsection of results, it mentions a β -sheet promoting valine. Is there any significance to this other than helix-breaking? Could it contribute to aggregation and formation of amyloids perhaps? This is not discussed further in the manuscript.

We used ThT assay to compare the fibrillation tendency between Msi-1C and Msi-2C. However, both proteins have no response to this assay (see inserted figure; TDP-43 is a positive control recorded in parallel). To find another probe to detect their amyloid formation, or the role that this valine may play, is currently out of our scope.

7. Rephrase "residue stretches" in the first sentence of the third subsection of results.

We have changed to "additional regions".

8. Consider moving the FRAP data from Supp. Fig. 6 to the main text since it seems to be a key result that Musashi-1 with the helical region forms aggregates or dynamically-arrested condensates.

We have merged Supplementary Fig 6a to Fig 4. In combination with replying to major comment #6, we have also moved Supp. Fig 6b to Fig. 4. Therefore, the original Supp. Fig 6 is not in the revised version.

Reviewer #2 (Remarks to the Author):

This article analyzed the IDR regions of Musashi protein family from the aspect of molecular biology, molecular evolution, and bioinformatic. The results are interesting and can contribute the elucidations of functions of IDRs. Although I am positive about accepting the article, I would like the authors to consider the following points.

We thank the reviewer's positive opinion on our work.

1) I cannot agree the section of "a-helicity is commonly involved in promoting self-assembly" in p8. It is unclear what the helical score, however, it seems multiplying a helical propensity show in Fig.5a of a site in interest and that of the previous residue. You can easily find the helix propensity by using the AAindex database (<https://www.genome.jp/aaindex/>, search with helix or something), and that of A and E of the classical parameter of "Chou and Fasman" are 1.42 and 1.51, and the other parameters recently published show similar. Then, the parameters shown in Fig5.a cannot be accepted. Since the state-of-the art secondary structure prediction programs such as psi-pred and others have accuracy in the practical use, I recommend using one of these programs to conduct the proteome wide analysis.

Based on the reviewer's suggestion, we have used RaptorX (the program we used to estimate helicity in Figure 6) to predict the helicity of all proteins' IDRs (longer than 40 a.a. in a row). The results show a similar trend. We have now put this new analysis as Figure 5d. We have also revised the paragraph discussing this point: ".....We also used RaptorX to predict the residues with a-helical propensity among the IDRs of the human proteome, RBPs, and mRBPs. We set arbitrary criteria (all other criteria show the same trend) to count those IDRs with at least five consecutive residues predicted higher than 0.8 (1.0 is the full scale) a-helical propensity in the program. Fig. 5d shows the same tendency as the polyalanine score: mRBPs' or RBPs' IDRs are more likely to have a-helical elements within them. In these analyses,....." We have added a new paragraph in the Materials and Methods to describe how we did the analysis and have deposited these scripts to Github (in the "Data Availability" statement).

However, we would also like to keep our original analysis. Our original aim is to show that "polyalanine" frequently appears in the RBPs' IDRs. To have more polyalanine counts which may be potentially disrupted by met or lue, we designed a set of tentative scores based on this and other studies (e.g. TDP-43). We apologize that we misused the terms between "a-helix" and "polyalanine". We mean "polyalanine" in many sentences instead of "a-helix". Although polyalanine in an IDR would form a-helix (as shown in this study and many others), a-helical elements within IDRs do not necessarily comprise

alanine. We thank the reviewer for pointing this ambiguity out, and we have now corrected many terms of “a-helix” to “polyalanine”, including in the figure and its legend. Both analyses lead to the same conclusion: “We attribute the results to that RBPs often join biomolecular condensates for their functions, and polyalanine/a-helix is one feature of IDRs that contributes to self-assembly.”

2) The clouds of Fig5c are not visible, and what is mRBP. I cannot find this term before P8 and no explanation is found.

We have changed the dots for random-selection results to black for clarity. We’ve also added “.....We also analyzed a group of 692 mRNA binding proteins (mRBPs) because many reported RBPs with LLPS-related functions reside in this category.....” to explain mRBP and why we analyzed them.

3) It is better to have organism labels in the alignments in SupFig.2.

They are now labeled in the revised Supplementary Fig. 2.

Reviewer #3 (Remarks to the Author):

Comment:

Shih-Hui Chiu and co-authors in their current work have focused on how IDRs evolve in protein paralogs and how the structured segments of the protein buffer the variational properties inflicted by the disordered segments in the protein. **This is a very interesting and relevant question in the context of the cellular unfoldome and how it evolves.** The authors in their test case have worked on Musashi-1 paralogs and have shown lower prion-like tendency of Musashi-1’s IDRs compared to that in Musashi-2’s and it is compensated by its higher alpha-helical propensity to assist their assembly. The problem/question is introduced with an interesting and clinically relevant example. This puts the point that protein paralogs in general have been studied but not in the context of disordered systems (IDP).

Available literature talks about the paralog proteins in general. This paper talks about the contribution of IDRs in modulating the behavior of the paralogs and how it impacts physico-chemical properties across the scales including condensate forming tendency. **This is an interesting read from the perspectives of the physics of the disordered systems. The article is well composed and written in a way so that the message can reach out to more general readership.** The article can be considered for publication but there are certain things which needs to be addressed.

We thank the reviewer for the high praise of our work. We also appreciate the constructive or inspiring suggestions.

Points and Suggestions:

1. Although the authors have explained the biological features associated with Masashi protein family, it would be better if they can put **this in the perspective of other proteins**. In the discussion section the authors have compared Masashi family traits with other proteins, but it would be better to **put some of this information in the introduction section as well**. This would make things clearer for a broader readership and to disordered systems physics community.

We have added, “We also compare the IDRs’ properties of other well-studied RBPs related to Musashi proteins, including TDP-43 and hnRNP A1.” at the end of the Introduction. We hope this sentence can bring interest to more general readers.

2. RNA molecules inherently have their own secondary structures. To interact RNA binding proteins sometimes need to utilize their less structured segment to have conformational complementarity. It would be great if this relation is discussed in some of the segments.

We are aware that RNA molecules are also involved in LLPS (for example, <https://pubmed.ncbi.nlm.nih.gov/26412307/>). In most cases, the RNA recognition motifs (RRMs) bind specific RNA sequence/structure and use their IDRs as addition “cross-link” to provide multivalency for LLPS (see the Graphic Abstract of the above reference for example). However, because we only focus on the properties of IDRs and their relation to evolution, we think discussing the conformational complementary to RNA molecules is not directly related to our current study.

3. Although authors in the discussion have talked about other proteins, especially in the context of liquid-liquid-phase separation and condensate forming tendency, a more detailed comparisons with some other model protein as parallel case study would have been interesting and would have strengthened the concept.

To the best of our knowledge, no study has reported experimental results on how IDRs evolves different traits for convergent or divergent functions through the lens of structural biology and biophysics. Therefore, the difference of IDR properties in Musashi paralogs is the novelty of this study. We believe there will be many other examples for this community to explore. Therefore, we conclude with the sentence that “.....our work could be used as a template to investigate IDRs in other paralogs and how their functions have diversified or been preserved during evolution.”

4. In the abstract authors have used the phrase "Without folded structure to restrain": Does that mean how much of the protein is folded? It would be interesting to look at the type of fold or super-secondary structures inside the folded segment are mostly evolutionarily selected for these kinds of proteins.

About one-third of eukaryotic proteins have intrinsically disordered regions. We are trying to say that the folded part of a protein is more conserved than its intrinsically disordered counterpart. See the inserted figure for an example (from <https://pubmed.ncbi.nlm.nih.gov/32144274/>). The structured regions (C-terminal half in this case) are more conserved (highlighted in orange colors) than the IDRs (N-terminal regions). This observation is a general trend that folded-domains evolve slower than the IDRs as shown in another manuscript under revision (see Fig. 2 in <https://www.biorxiv.org/content/10.1101/2021.10.28.466204v1>).

In this Musashi article, we want to address that no matter how diverse the IDRs can evolve, they may have converged to similar functions (e.g. LLPS in this manuscript). We think the evolution rate between the folded and unfolded regions are not correlated. At the beginning of the Discussion, we addressed “Without the constraints of a fixed shape, the IDRs in a paralog family can evolve more freely than structured regions, either gaining new functions or compensating for lost ones.” We have also revised the Abstract (also based on reviewer #1’s suggestion) as “Without a folded structure to restrain them, IDRs mutate more diversely along with evolution. However, how the diversity of IDRs in a paralog family affects their functions is unexplored.” We hope these explanations make our point clearer.

5. It would be interesting if the sequences are scanned for slow codons or kinetic traps and the paralogs are analyzed in the context of kinetics of folding.

The kinetics of folding is not relevant to our study because the IDPs do not fold. However, we think the reviewer brought up an interesting perspective from codon usage. The codon usage affects translation speed and thus also has been “selected” during evolution (e.g. <https://pubmed.ncbi.nlm.nih.gov/26849192/>). The prion-like domain may have evolved using the rare codons to slow down the translational rate to prevent

aggregation; we guess it would be similar to the kinetic traps for structural proteins.

Investigating the codon usage is an interesting point for the paralogs of IDPs. However, in our experimental approaches, the proteins' IDRs were purified, and thus what codons they use and how fast/slow they are translated are not related to our works and have no effect on our conclusion. We think additional discussion on codon usage also distracts from the main message we want to deliver. Investigating from the DNA/RNA level is indeed a very interesting perspective of looking at IDRs paralogs. We believe this should be an independent work.

6. Did the authors look for residue positions in the sequence space in the context of stabilizing/destabilizing mutations by deploying some deep mutational scans? That would be interesting in the context of IDRs as inherently they should have higher mutational tolerance owing to low complexity in structures. But on the other hand, their functional role would not allow for few select set of mutational events. A deep mutational analysis would be interesting in this context.

Further I recommend carrying out co-variational analysis to capture the dynamics of sequence variations in the IDR regions and how that correlate to pair-wise linked positions in the protein structure.

We are aware that TDP-43 has been done in a similar approach (<https://www.nature.com/articles/s41467-019-12101-z>), which should be also applicable to these two Musashi proteins. However, deep mutational scanning requires extensive work, and we think this is out of the scope of the article. Alternatively, evolution might have already provided hints on mutational tolerance. We have compared the orthologs of all human RBPs and are trying to find conserved traits (<https://www.biorxiv.org/content/10.1101/2021.10.28.466204v1>). Meanwhile, we also find the limitations of the current multiple sequence alignment approach because of the often-missing segments in IDRs. We are currently trying to use machine-learning approaches for all orthologs of proteins, and we hope this approach will provide traits of IDRs and their functional links. We believe this would be an alternative and probably more efficient approach to investigate mutational tolerance.

We also think discussing the dynamics of sequence variation in the IDRs of Musashi proteins or how they correlate to their structure is not relevant to our current study. Here, we aim to convey the concept that IDRs can evolve different physical properties for the same function. We think these suggested

approaches, although they are potentially interesting projects by themselves, would not add more information to our current conclusion.

7. While comparing Musashi-1 with other proteins in the context of LLPS authors must make things clearer. This section in the discussion forms an important component of the paper and contains some key messages.

We thank the reviewer for this suggestion. We have largely rephrased sentences in the final section and have added passage sentences: Including the opening of defining our scope: “According to the RBP census study, many other RBPs were grouped in the paralog family of Musashi proteins based on sequence similarity. We confirmed this by searching for....”; some passages to smooth the logic flow: “.....agreeing with the previous study. Accordingly, we compared the IDRs’ properties of these Musashi protein’s “distant relatives”: Studies of.....” and “.....Although the IDRs of these RBPs have different properties, they undergo LLPS.....”. We have also rearranged the final sentences: “As a conclusion, we suggest that IDRs evolve whatever traits are beneficial to function, which accords with François Jacob’s statement that “evolution does not produce novelties from scratch”. He was referring to the diversity of all lifeforms but his words also apply to IDRs, which acquire new functions through evolutionary “tinkering” between prion-likeness, α -helicity, aromatic residues, etc. Our work could be used as a template to investigate IDRs in other paralogs and how their functions have diversified or been preserved during evolution.” We hope the revised version has a more coherent logic flow and delivers clearer messages.

8. It would be interesting if authors touch upon the sequence variations in the IDR segments of the paralogs in the context of **protein quality control**.

This comment is also an interesting idea. The IDRs paralogs may have roles in protein quality control, not in the sense of helping protein fold (because IDRs have no structure), but the removal of aggregated or overexpressed proteins. For example, different levels of accessibility for ubiquitylation in the IDRs might have evolved to control the protein level or quality. However, we think discussing protein quality control is also not relevant to the main message we want to deliver.

REVIEWERS' COMMENTS:

Reviewer #1 (Remarks to the Author):

The authors have done a good job addressing my concerns. I only have very minor issues (related to my previous comments) in response to the draft in its current state.

minor point 5: While only a minor comment, I still think it would be helpful to consolidate Fig. 2a to just one panel. Having the same data set visualized twice may confuse the reader. Alternatively, the first data set could be removed from the second panel.

minor point 6: Since β -sheet content is outside the scope of this study, I would suggest removing " β -sheet-promoting" from before valine.

This is a very nice study and believe it would be a good addition to the literature.

Reviewer #3 (Remarks to the Author):

Authors have carefully gone through all the recommendations and points suggested. They have addressed majority of the points and rephrased many of the sections in the article accordingly. Further in the rebuttal letter they have acknowledged points which are interesting but little beyond the scope of the current work. The rephrased sections provide clarity to the inferences drawn. With all the modifications I think the article would be an interesting read for the community. I recommend the article for acceptance in its current form.

REVIEWERS' COMMENTS:

Reviewer #1 (Remarks to the Author):

The authors have done a good job addressing my concerns. I only have very minor issues (related to my previous comments) in response to the draft in its current state.

minor point 5: While only a minor comment, I still think it would be helpful to consolidate Fig. 2a to just one panel. Having the same data set visualized twice may confuse the reader. Alternatively, the first data set could be removed from the second panel.

minor point 6: Since β -sheet content is outside the scope of this study, I would suggest removing " β -sheet-promoting" from before valine.

This is a very nice study and believe it would be a good addition to the literature.

We thank the reviewer's comment. We have removed the first data set in the right panel of Fig.2a and the phrase " β -sheet-promoting".

Reviewer #3 (Remarks to the Author):

Authors have carefully gone through all the recommendations and points suggested. They have addressed majority of the points and rephrased many of the sections in the article accordingly. Further in the rebuttal letter they have acknowledged points which are interesting but little beyond the scope of the current work. The rephrased sections provide clarity to the inferences drawn. With all the modifications I think the article would be an interesting read for the community. I recommend the article for acceptance in its current form.

We thank the reviewer for thought high of our work.